# Assessing the Various Antagonistic Mechanisms of *Trichoderma* Strains against the Brown Root Rot Pathogen *Pyrrhoderma noxium* Infecting Heritage Fig Trees

**DOI:** 10.3390/jof8101105

**Published:** 2022-10-19

**Authors:** Harrchun Panchalingam, Daniel Powell, Cherrihan Adra, Keith Foster, Russell Tomlin, Bonnie L. Quigley, Sharon Nyari, R. Andrew Hayes, Alison Shapcott, D. İpek Kurtböke

**Affiliations:** 1School of Science, Technology and Engineering, The University of the Sunshine Coast, 90 Sippy Downs Dr, Sippy Downs, QLD 4556, Australia; 2Brisbane City Council, Program, Planning and Integration, Brisbane Square, Level 10, 266 George Street, Brisbane, QLD 4000, Australia; 3Forest Industries Research Centre, The University of the Sunshine Coast, 90 Sippy Downs Dr, Sippy Downs, QLD 4556, Australia

**Keywords:** *Trichoderma*, *Pyrrhoderma noxium*, antagonism, mycoparasitism, biological control, heritage fig trees, *Ficus macrophylla*

## Abstract

A wide range of phytopathogenic fungi exist causing various plant diseases, which can lead to devastating economic, environmental, and social impacts on a global scale. One such fungus is *Pyrrhoderma noxium*, causing brown root rot disease in over 200 plant species of a variety of life forms mostly in the tropical and subtropical regions of the globe. The aim of this study was to discover the antagonistic abilities of two *Trichoderma* strains (#5001 and #5029) found to be closely related to *Trichoderma reesei* against *P. noxium*. The mycoparasitic mechanism of these *Trichoderma* strains against *P. noxium* involved coiling around the hyphae of the pathogen and producing appressorium like structures. Furthermore, a gene expression study identified an induced expression of the biological control activity associated genes in *Trichoderma* strains during the interaction with the pathogen. In addition, volatile and diffusible antifungal compounds produced by the *Trichoderma* strains were also effective in inhibiting the growth of the pathogen. The ability to produce Indole-3-acetic acid (IAA), siderophores and the volatile compounds related to plant growth promotion were also identified as added benefits to the performance of these *Trichoderma* strains as biological control agents. Overall, these results show promise for the possibility of using the *Trichoderma* strains as potential biological control agents to protect *P. noxium* infected trees as well as preventing new infections.

## 1. Introduction

Fungi are the most dominant group of plant pathogens, causing some of the world’s most serious plant diseases on a wide range of crops and pose a continuous threat to global food security [1,2,3,4]. One such fungus is *Pyrrhoderma noxium* which causes the destructive plant disease known as brown root rot in tropical and sub-tropical regions, such as Africa, Asia, Central America, the Caribbean, Australia and Oceania. In the case of Australia, sub-tropical conditions in the state of Queensland (QLD) have been reported to facilitate the establishment of this disease [5]. Its highly invasive nature has already killed many significant trees including several heritage fig trees found in the public spaces of Brisbane city, the capital of QLD [5]. Infected trees in urban parks, along the roads and power lines are imposing a great risk to surrounding people and properties. This necessitated the need to identify suitable disease control measures to manage its spread.

One widely used strategy to manage fungal plant diseases is the utilisation of chemical fungicides. However, chemical-based control measures (fungicides and fumigants) are not environmentally friendly and the chemical residues can build-up in the food chain and cause negative effects on human health [5,6,7]. In recent years, the role of biological control agents (BCAs) has been getting more attention as alternative control measures over hazardous chemicals. Various *Trichoderma* species have been found to play important roles in managing many phytopathogenic fungi [8,9,10,11,12,13].

The antagonism by *Trichoderma* is proposed as a combination of direct and indirect mechanisms [14]. Some of these include the recognition of host signals by *Trichoderma* which appears to be an essential step for successful continuation of the mycoparasitism against pathogenic fungi [15]. Additionally, during the initial stages of direct mycoparasitism, *Trichoderma* coil around the pathogenic fungal hyphae or produces hook-shaped and appressorium-like structures [16,17]. Previous research has suggested that the coiling is mediated by the signals generated from the host, its surface structure, or the by-products of its initial degradation by *Trichoderma* [18,19].

Mechanical and enzymatic penetration of the pathogen by the *Trichoderma* are the next step of interaction. The enzymes, mainly chitinase (*chit*), β-glucanase (*bgn*) and protease (*pra* and *prb*) produced through the expression of biological control associated genes are involved in this step [20,21,22,23,24,25]. The expression of *chit*, *bgn*, *pra* and *prb* genes in *Trichoderma* were reported to be induced after physical contact between *Trichoderma* and the pathogen [26,27]. The cell wall degrading enzymes produced by the *Trichoderma* act synergistically and digest the host cell wall at the interaction site before penetration [28]. *Trichoderma* with the genes encoding chitinase, endo chitinase, β-1,3-glucanase and proteases were reported to have much stronger antifungal activity against a wide range of phytopathogens [24,26,27,29,30,31,32]. While the *qid* gene in *Trichoderma* is reported to be associated with the synthesis of cysteine-rich cell wall protein, cellular protection, adherence to hydrophobic surfaces and formation of specialised structures such as appressoria [23,33]. This gene was reported to be induced both before and after physical contact with the pathogen [26,27].

In addition, volatile and non-volatile compounds produced by *Trichoderma* are also capable of inhibiting the growth of pathogens [34]. A wide range of bioactive secondary metabolites produced by *Trichoderma* species are reported to show inhibitory effect on various phytopathogenic fungi including *Alternaria alternata*, *Botrytis cinerea*, *Fusarium* species, *Pythium* species, *Rhizoctonia solani*, *Sclerotinia sclerotiorum*, and *Ustilago maydis* [35,36,37]. Mainly, pyrones, koninginins, viridins, nitrogen heterocyclic compounds, azaphilones, butenolides, hydroxy-lactones, diketopiperazines, isocyano metabolites and peptaibols compounds produced by *Trichoderma* species were reported to exhibit bioactivity against many phytopathogenic fungi [38,39,40,41,42,43,44,45,46,47,48]. The VOCs of *Trichoderma* species associated with antifungal ability, induce a defence response in plant and promote plant growth [49,50,51,52,53,54]. Volatile compounds (VOCs) released by the *Trichoderma* species were reported to directly inhibit the growth of pathogens or cause abnormal changes of pathogens [49,55]. The *Trichoderma* VOCs belong to various structural classes including mono-and sesquiterpenes, lactones, esters, alcohols, ketones or C8 compounds [56]. The alcohol compounds, mainly 3-methyl-1-butanol, 2-methyl-1-propanol and ethanol have been previously reported to be produced by different *Trichoderma* species [34,55,57,58].

Indirect mechanisms include competition for nutrients and space, plant growth promotion and induced systemic resistance to the plant [34,59]. The production of plant regulators by *Trichoderma* is an important mechanism often associated with stimulation of plant growth [60]. Previously conducted studies confirmed the IAA production ability of *Trichoderma* species including in *T. asperellum*, *T. harzianum*, *T. pinnatum*, *T. longibrachiatum*, *T. asperelloides*, *T. virens*, *T. atrobrunneum*, *T. guizhouense*, *T. paratroviride*, and *T. simmonsii* [61,62,63,64]. On the other hand, IAA produced by *Trichoderma* may cause physiological changes in the plant, such as cell wall-loosening and the release of nutrients. Subsequently assist the colonization of beneficial microorganisms [65,66]. These beneficial microorganisms can also compete with pathogens and limit their growth.

Iron is an essential component of various proteins and pigments for pathogenic fungi [67]. *Trichoderma* species produce a variety of siderophores, mainly related to the class hydroxamate to chelate the insoluble iron and make them unavailable to the pathogen [37,68]. Certain antagonistic strains of *T. asperellum*, *T. atrobrunneum*, *T. atroviride*, *T. gamsii*, *T. hamatum*, *T. harzianum*, *T. polysporum*, *T. reesei*, *T. virens*, *T. paratroviride*, *T. pyramidale*, *T. rufobrunneum*, *T. thermophilum*, *T. viridulum*, *T. guizhouense* and *T. simmonsii* were reported for siderophore production [37,63,69,70].

Thus, studying the antagonistic mechanisms of *Trichoderma* species against the pathogen in vitro gives insight into its effectivity as a biological control agent for environmental applications. Accordingly, two indigenous *Trichoderma* strains (#5001 and #5029) were tested to: (1) evaluate the antagonistic mechanisms of the strains against *P. noxium* strains; (2) study the expression of the biological control associated genes during their confrontation with *P. noxium*; (3) characterise the antifungal activity of volatile compounds emitted from these *Trichoderma* strains; (4) study the plant growth promoting traits of these *Trichoderma* strains which may be associated with plant growth promotion. Identification of effective biological control agent would protect the economically and culturally significant plants from *P. noxium* infection and also minimise the application of hazardous chemical control measures against this pathogen.

## 2. Materials and Methods

### 2.1. Genome Sequencing for the Molecular Identification of Trichoderma Strains

For the molecular identification, the *Trichoderma* strains (#5001 and #5029) were cultured on Potato Dextrose Agar (PDA, Oxoid, Hampshire, UK) and incubated at 28 °C for 5 days prior to the DNA extraction. DNA extraction was performed using DNeasy Plant mini Kit (Qiagen, Hilden, Germany) and following the manufacturer’s instructions. DNA samples were then sent to the Australian Genome Research Facility (AGRF) in Brisbane, QLD, for whole genome sequencing on NovaSeq 6000 platform. The data generated with the Illumina bcl2fastq pipeline version 2.20.0.422 were used in the downstream analysis.

#### 2.1.1. Draft Genome Assembly

Sequencing was checked for quality using FastQC v0.11.9 (https://www.bioinformatics.babraham.ac.uk/projects/fastqc/; accessed on 15 January 2022). Trimmomatic 0.39, multithreaded command line tool was used for removing adapter sequences and the low quality reads were filtered out [71]. SPAdes 3.15.3, assembler was used for creating the genome assemblies with K-mer sizes of 21, 25, 33, 55, 77, 91 and 101 [72]. Completeness of the *Trichoderma* draft genome assembly was evaluated using BUSCO v. 5.1.3 against the asomycota database [73]. K-mer depth was calculated using Jellyfish version 2 [74] and K-mer-based estimates of genome size was generated using GenomeScope [75].

#### 2.1.2. Assembly Preparation for Downstream Application

De novo genome annotation of both *Trichoderma* genomes were performed using GeneMark-ES [76] and AUGUSTUS [77] in Funannotate (v1.7.4) gene prediction pipeline (https://funannotate.readthedocs.io/en/latest/ accessed on 15 January 2022). Then, validated by mapping the RNAseq data of the reference strain *T. reesei* (PRJNA15571) to the genes to support the predictions. The newly annotated gene models were evaluated for completeness using BUSCO v5 in protein mode against the ascomycota_odb10 database.

#### 2.1.3. Phylogenetic Tree

For the identification of the closest match of the *Trichoderma* strains # 5001 and 5029, protein sequences of 12 *Trichoderma* strains were analysed together with predicted protein sequences of *Trichoderma* strains # 5001 and #5029. The protein sequences of *T. reesei* (GCF 000167675.1), *T. cornu-damae* (GCA 020631695.1), *T. arundinaceum* (GCA 003012105.1), *T. asperellum* (GCF 003025105.1), *T. atroviride* (GCF 000171015.1), *T. gamsii* (GCF 001481775.2), *T. virens* (GCF 000170995.1), *T. harzianum* (GCF 003025095.1), *T. lentiforme* (GCA 011066345.1), *T. simmonsii* (GCA 019565615.1), *T. guizhouense* (GCA 002022785.1), *T. semiorbis* (GCA 020045945.2) and protein sequence of *A. niger* (GCF 000002855.3) as out group were downloaded from the GenBank database (as of December 2021). Protein sequences were analysed with OrthoFinder v. 2.3.8, which grouped them into orthogroups [78,79]. Gene families that contained only one gene for each species (single copy orthologs) were selected to construct a phylogenetic tree using FastTree program integrated in OrthoFinder [80]. The phylogenetic trees of the *Trichoderma* isolates were constructed by maximum likelihood method [81]. Finally, the tree was visualized using iTOL [82].

### 2.2. Microscopic Investigation of the Mycelial Interaction between P. noxium and Trichoderma Strains

The mycelial interaction between *Trichoderma* strains and *P. noxium* was observed under an automated upright microscope system (LEICA DM5500 B) [83]. The *P. noxium* plug was inoculated at 0.5 cm distance from a coverslip embedded into the PDA at a 45° angle. After two days of incubation, a plug of the strains #5001 and #5029 (separately) was placed 1 cm away from *P. noxium* on the plate and incubated for 2 days at 28 °C. Once both fungi were grown onto the coverslip, fungal mycelia were stained with lactophenol cotton blue and observed for the mycoparasitism-related structures produced by *Trichoderma* strains, e.g., coiling and appressorium-like structures.

### 2.3. Detection of the Antagonistic Activities of Trichoderma Strains against P. noxium Using a Dual Culture Assay

The antagonistic activity of *Trichoderma* strains (#5001 and #5029) was tested against five different *P. noxium* strains: A, B, D, E and F using a dual culture assay [84]. *Trichoderma* and *P. noxium* plugs were inoculated on a PDA plate away from each other [84]. Due to the slow growth rate, *P. noxium* was inoculated two days before the inoculation of *Trichoderma* strains. After two days, a *Trichoderma* plug was placed 2 cm from the edge of the *P. noxium* colony. As a control, the pathogen and the biological control agents (BCAs) were placed in a similar manner on a fresh PDA plate without reciprocal fungal inoculations and incubated at 28 °C for 5 days. All pairings were carried out in quadruplicate. Growth inhibition percentage for tested phytopathogenic fungi was determined [84].
$$\mathrm{Inhibition\%}=\left[\frac{\mathrm{Colony}\;\mathrm{diameter}\;\mathrm{of}\;P.\;noxium\;\left(\mathrm{control}\right)-\;\mathrm{Colony}\;\mathrm{diameter}\;\mathrm{of}\;P.noxium\;\left(\mathrm{treatment}\right)}{\mathrm{Colony}\;\mathrm{diameter}\;\mathrm{of}\;P.\;noxium\;\left(\mathrm{control}\right)}\right]*100$$

### 2.4. Assessing the Inhibitory Effect of Volatile Compounds Produced by Trichoderma on P. noxium Growth

To study the effect of volatile compounds of *Trichoderma* strains on *P. noxium* (A, B, D, E and F), the sandwich plate method described by Dennis and Webster, 1971b [85], was used. A *Trichoderma* plug (3 days old) was cultured on PDA medium and incubated for 4 days at 28 °C. After that, the lids of the *Trichoderma* culture plates were replaced by the PDA plates (bottom part) freshly inoculated with a *P. noxium* plug, sealed with three layers of parafilm, and incubated at 28 °C for 3 days. Controls of *P. noxium* were prepared by sandwiching with an un-inoculated PDA plate. All pairings were carried out in quadruplicate. Effect of volatile compounds produced by BCAs on the pathogen was evaluated by comparing the growth area of *P. noxium* in the treatment plates with the control plates [85].

### 2.5. Changes in the Volatile Compounds Produced by Trichoderma Strains before and after the Confrontation with P. noxium

To identify the volatile compounds produced by the pure cultures of *Trichoderma* strains (#5001 and #5029), each strain was prepared by placing 2 mm agar plugs in the center of a headspace vial (20 mL) containing PDA medium. To study the changes in the volatile compounds produced by *Trichoderma* strains during the interaction with pathogen, co-cultures were prepared by placing two 2 mm agar plugs of the strains #5001 and 5029 (separately) and *P. noxium* (strain B was used as representative of *P. noxium*) at the center of the headspace vial and incubating for 7 days at 28 °C [86]. Due to the slow growth rate, *P. noxium* was inoculated one day before the inoculation of *Trichoderma*. Pure cultures of *P. noxium* were also prepared in headspace vials and used for identifying the volatile compounds produced by *P. noxium* in the co-culture setup.

Uninoculated PDA medium was used as blank [86]. All the inoculated vials and controls were incubated for 7 days at 28 °C. Volatiles in the headspace of the above cultures were analyzed using Solid Phase Microextraction (SPME). Fibers (75 µm Carboxen/PDMS, Merck) were inserted in the headspace vial, and exposed for 15 min. Volatiles were desorbed from the fiber when it was injected into the inlet port and analyzed with a gas chromatograph (GC) (Agilent 6890 Series) coupled to a mass spectrometer (MS) (Agilent 5975) and fitted with a silica capillary column (Agilent, model HP5-MS, 30 m × 250 µm ID × 0.25 µm film thickness). Gas chromatograph conditions for acquiring data were—inlet temperature: 250 °C, carrier gas: helium at 51 cm/s, split ratio 13:1, transfer-line temperature: 280 °C, initial temperature: 40 °C, initial time: 2 min, rate: 10 °C/min, final temperature: 260 °C, final time: 6 min. The MS was held at 280 °C in the ion source and the scan rate was 4.45 scans/s. Tentative identities were assigned to peaks with respect to the National Institute of Standards and Technology mass spectral library. Mass spectra of peaks from different samples with the same retention time were compared to ensure that the compounds were indeed the same. Negative control results were compared with volatiles from blank PDA control and common substances were removed during interpretation of results.

### 2.6. Effect of Diffusible Compounds Produced by Trichoderma Strains on P. noxium

To examine the effects of diffusible (non-volatile) compounds produced by the strains # 5001 and #5029 on *P. noxium* growth, biological control agents and pathogen (*P. noxium* strain B) were co-cultured on PDA and incubated at 28 °C for 7 days. After the incubation period, culture medium was cut into small pieces and the diffusible compounds were extracted using ethyl acetate [87]. Three extractions were performed at 24-h intervals on a shaker at 120 rpm at 20 °C. All the extracts were filtered through a Whatman No. #1 filter paper [87]. The ethyl acetate extracts were concentrated on a rotary evaporator at room temperature and further evaporated using Genevac. The dried crude extracts were measured and dissolved in appropriate volume of ethyl acetate to get the concentrations of 500, 1000, 1500, 2500, 5000 and 10,000 ppm and used for fungicidal assay using paper disc diffusion technique [88].

To determine the fungicidal activity of the crude extract against *P. noxium*, an agar plug from a 3-day-old mycelial culture of *P. noxium* strain B was inoculated at the center of a fresh PDA plate. Two sterile disks were kept both sides of the inoculated plug at 2.5 cm distance. One of the disks was saturated with 100 µL of the crude extract and 10 µL of Dimethyl sulfoxide (DMSO) and the other disk was saturated with solvents (100 µL ethyl acetate and 10 µL DMSO) and used as the control. The plates were incubated at 28 °C for 5 days. The fungicidal activity of crude extract at different concentrations was assessed in terms of the area of fungal inhibition and expressed as the percentage of the inhibition of fungal growth as mentioned above. The fungicidal assay was performed with three replicates.

### 2.7. The Biocontrol-Associated Genes Expression in Trichoderma Strains during the Interaction with P. noxium

In order to study changes in the expression level of biological control related genes in *Trichoderma* strains (#5001 and #5029), a confrontation assay was conducted using *P. noxium* [22]. The strain B was used as the *P. noxium* representative in this study. Six biological control associated genes (chitinase 33 (*chit*), endochitinase 42 (*endo*), β-1,3-endoglucanase (*bgn*), trypsin-like protease pra1 (*pra*), subtilisin-like protease prb1 (*prb*) and QID74 protein (*qid*)) previously reported in *Trichoderma* spp. [27] for the expression upon pathogen confrontation were studied during the interaction between *Trichoderma* strains (#5001 and #5029) and *P. noxium* strain B. PDA plates were overlaid with cellophane and were inoculated with agar plaques of the *P. noxium* strain B and the strains # 5001 or #5029 (separately) as explained above and incubated for 3 days at 28 °C. The membrane was used to facilitate the removal of the *Trichoderma* mycelia at the contact point from the plate for RNA extraction [22,24]. For the controls, *Trichoderma* strains (#5001 and 5029) were grown alone on PDA plates at the same growth conditions, while the peripheral hyphae were used for RNA extraction.

For RNA extraction, mycelia (90 mg) of the *Trichoderma* strains #5001 and #5029 from the treatment and control plates were immediately placed in Eppendorf tubes with Zirconia silica beads (few of the 23 mm beads and a small quantity of 0.1 mm diameter beads) and frozen in liquid nitrogen for 30 s and then placed into a Qiagen TissueLyser adapter for efficient grinding for 40 s at 23,000 rpm. The liquid nitrogen freezing and grinding steps were repeated twice and the adapter positions were reversed in the TissueLyser each time. The ground mycelia were used for total RNA extraction using RNeasy Plant Mini Kit (Qiagen, Hilden, Germany) as per manufacturer’s instructions. NanoDrop was used for RNA quantification.

Genomic DNA elimination step followed by cDNA synthesis was carried out using the reaction components provided in the QuantiTect^®^ Reverse Transcription Kit (Qiagen, Hilden, Germany) and following manufacturer’s instructions. The cDNA concentration was quantified using a NanoDrop. The changes in the expression level of biological control associated genes of *Trichoderma* during the confrontation with *P. noxium* was analysed using qPCR techniques. This method is widely used for the quantification of gene expression [22,24]. Biological control associated genes in strains #5001 and #5029 were identified via homology sequence search against the biological control genes of *T. harzianum* CBS 226.95 (V1.0) previously reported [24]. The mRNA of *T. harzianum* CBS 226.95 (V1.0) were obtained from Genbank (https://www.ncbi.nlm.nih.gov/genbank/ as of 15 December 2021) and the gene transcripts (*chit*, *endo*, *bgn*, *pra*, *prb* and *qid*) were identified via locating the specific primer sequences reported [24]. The identified transcripts of *T. harzianum* were individually BLAST against the blast library of mRNA sequences from the strains #5001 and #5029 created in Qiagen CLC Genomics Workbench version 20.0. The homology sequences (with lowest E-value and highest percentage of identity) in the transcript of *Trichoderma* strains #5001 and #5029 were selected for primer designing. The specific primers were designed using Primer 3 (http://primer3.ut.ee/ accessed on 15 January 2022) (Appendix A) and manufactured by Merk (Australia). Quantitative real-time PCR was used for amplifying the transcripts from individual genes using QuantiTect SYBR Green PCR Kit (Qiagen, Hilden, Germany). Each reaction consisted of 10 μL QuantiTect SYBR Green PCR master mix, 1 μL each of forward and reverse primers (final concentration of 0.5 μM), 2 μL cDNA, and 6 μL Rnase-free water. qPCR conditions were: one cycle of initial denaturation step at 95 °C for 15 min to activate the HotStarTaq DNA Polymerase, followed by 35 cycles of denaturation at 95 °C for 15 s, annealing at 59 °C for 30 s and extension at 72 °C for 30 s. High resolution melting (HRM) was carried out over the ranging from 65 °C to 95 °C with temperature increasing steps of 0.5 °C every 4 s. RNase-free water was used in non-template control [24].

Genes encoding elongation factor 1-α and β-actin were used as internal references for normalization of gene expression in *Trichoderma* [24,89]. Relative expression levels of individual genes were calculated from the threshold cycle using the Pfaffl method for efficiency correction [90]. A standard curve was generated to calculate the gene-specific PCR amplification efficiency and correlation coefficient for each gene [24].

### 2.8. Effect of Volatiles Produced by Trichoderma Strains on the Growth Parameters of Arabidopsis thaliana

The effect of VOCs of *Trichoderma* on plant growth parameters was studied using modified double plate-within-a-plate method described by Lee et al. 2016 [54]. *Arabidopsis thaliana* was used as model plant [54]. Two small Petri dishes (50 mm) with PDA were placed in a larger plate (140 mm) and 50 mL of Murashige and Skoog salt (MS) medium (pH 5.7) containing 3% sucrose and 0.7% agar was poured into the larger plate (Appendix A). *A. thaliana* seeds were surface sterilised for 5 min in 70% ethanol, soaked for 7 min in 10% sodium hypochlorite solution, and rinsed four times with distilled water. Finally, seeds were kept in distilled water and vernalized for 2 days at 4 °C in dark. Sterilised seeds were inoculated on MS media and incubated for 3 days. Ten germinated seeds selected from contamination free plates were reinoculated on the MS media portion established in the double plate-within-a-plate setup (Appendix A). In the meantime, small plates in the double plate-within-a-plate units were inoculated with 5 μL of a *Trichoderma* spore suspension (containing 1 × 10^6^ spores). For controls, plants were exposed to the PDA alone. The VOCs mediated plant growth promotion is often associated with exposure period [54]. Therefore, this study was conducted for a period of 12 days to provide enough volatile exposure to the plants. All the plates were sealed and incubated at 25 °C. The experiment was performed in a complete randomized block design with 2 treatments (*Trichoderma* strains #5001 and #5029) and 1 negative control with 5 replicates (corresponding to 10 plants/plate).

The effect of VOCs on the growth parameters of *A. thaliana* was evaluated by comparing the fresh mass, shoot length, root length and total chlorophyll content of plants exposed to *Trichoderma* strains with unexposed plants. Ten plates were pooled and biomass was measured (n = 5, corresponding to 50 plants). Total chlorophyll content of *A. thaliana* was determined spectrophotometrically [54]. Fifty milligrams of shoots collected from each treatment were ground and submerged overnight in 100 μL of 80 % acetone in the dark at 4 °C. The following day, absorbance readings were measured at 645 nm (A_645_) and 663 nm (A_663_) using an EnSpire™ multimode plate reader [52,54]. The total chlorophyll content was calculated from the following equation [(8.02)(A_663_) + (20.2)(A_645_)]V/1000 × W, where V is volume and W is plant fresh mass [52,54].

### 2.9. Indole 3-Acetic Acid Production by Trichoderma Strains

The IAA producing ability of *Trichoderma* strains was studied in potato dextrose broth (PDB, Oxoid, Hampshire, UK) supplemented with l-tryptophan [91]. Filter (0.2 μm membrane filter, Whatman) sterilised tryptophan (0.2 % w/v) was incorporated after sterilising the media at 15 psi at 121 °C for 15 min. During the preparation of 0.2% l-tryptophan aqueous solution, a few drops of NaOH (1 M) was used for increasing solubility of tryptophan. After adding the l-tryptophan, culture medium pH was adjusted to 5.6 with filter sterilized HCl (1 M).

Culture flasks containing 50 mL of PDB broth with l-tryptophan were inoculated with two agar plugs removed from a 3-day old *Trichoderma* culture plate and the flasks were incubated at 28 °C at 120 rpm [92,93]. Cell-free culture filtrates collected on day 7 were used for testing the indole 3-acetic acid production. IAA was measured by mixing 1 mL of the cell-free culture filtrate with 2 mL of Salkowski reagent (2% 0.5 M FeCl_3_ in 35% HClO_4_) [94]. The reaction mixture was incubated at room temperature for 20 min and the absorbance was recorded at 530 nm in a UV-Vis spectrophotometer [95]. IAA produced by the *Trichoderma* was determined from the standard curve. The experiment was performed in triplicate, IAA produced by the *Trichoderma* strains are presented as mean values with a standard deviation of three replicates.

### 2.10. Siderophore Production by Trichoderma Strains

The siderophore production by *Trichoderma* strains was studied on universal siderophore assay medium using chrome azurol S (CAS). The medium was prepared according to the procedure described by Schwyn and Neilands, 1987 [96] and Louden et al. 2011 [97]. A culture plug removed from 3-day old *Trichoderma* plate was placed at the centre of CAS agar medium and incubated at 28 °C for 7 days. After the incubation period, plates were observed for evidence (orange halo) of siderophore production.

### 2.11. Statistics

The effect of total volatiles produced by *Trichoderma* strains on the growth of different *P. noxium* strains (A, B, D, E and F) were analysed via one-way ANOVA followed by Tukey’s test. The differences in inhibitory effect caused by the *Trichoderma* strains # 5001 and #5029 were compared using the paired *t*-test. Similarly, the inhibitory effect displayed by the *Trichoderma* strains on the growth of different *P. noxium* strains during the confrontation was studied using one-way ANOVA followed by Tukey’s test. The differences in mycoparasitism caused by the *Trichoderma* strain # 5001 and #5029 were compared using the paired *t*-test. The effect of crude extract of *Trichoderma* strains on the growth of *P. noxium* strains B at different concentrations was analysed via one-way ANOVA followed by Tukey’s test. The differences in inhibitory effect caused by the crude extract of *Trichoderma* strains # 5001 and #5029 were compared using the paired *t*-test. The results from gene expression study were analysed using a Student’s *t*-test to compare the changes in expression levels between control and treatments. The effect of VOCs of *Trichoderma* strains on *A. thaliana* was compared between the strains and also with control via one-way ANOVA followed by Tukey’s test. The IAA and siderophore producing ability of the strains # 5001 and #5029 were compared using the paired *t*-test. All the statistical analyses were performed using SPSS statistics 26 [41].

## 3. Results

### 3.1. Molecular Identification of Trichoderma Strains via Whole Genome Sequencing

The Illumina sequencing generated 222 million, 150-bp paired-end reads for *Trichoderma* strain #5001 and 202 million, 150-bp paired-end reads for *Trichoderma* strain #5029. The genome of *Trichoderma* strain #5001 assembled in to 116 scaffolds with a total assembly length 32.126 Mb (N50 = 493.501 KB and N90 = 196.64 KB), a GC content of 53.33%, and a largest scaffold size of 1.473 Mb (Table 1). Based on Kmer-based genome size estimation size of the *Trichoderma* strain #5001 genome was 31,444,261 bp. In total, 8812 protein coding genes were predicted in the *Trichoderma* strain #5001 genome. The analysis showed the presence of 99% BUSCO completeness. *Trichoderma* strain #5029 genome consists of 85 scaffolds sequence with a total assembly length 32.132 Mb (N50 = 599.273 KB and N90 = 264.963 KB), a GC content of 53.33%, and a largest scaffold size of 1.76 Mb (Table 1). Genome size of *Trichoderma* strain #5029 was estimated as 31,502,229 bp. In total, 8969 protein coding genes were predicted in the *Trichoderma* strain #5029 genome. The analysis showed the presence of 99.06% BUSCO completeness. These assembly quality metrics indicate that we have achieved a high level of contiguity and completeness for both these genomes in comparison with other published assemblies.

The program OrthoFinder identified 158,696 gene clusters, from which 4653 orthologous genes were shared across all 15 *Trichoderma* species. From these shared gene clusters, 3059 orthologous genes present as single copies were used for the phylogenetic analysis. *Trichoderma* strain #5001 and #5029 form a distinct cluster with *T. reesei* (GCF 000167675.1) (Figure 1) indicating they are most closely related to *T. reesei.*

### 3.2. Microscopic Investigation of the Mycelial Interaction between P. noxium and Trichoderma Strains

During the interaction between *Trichoderma* strain #5001 (T) and *P. noxium* (P), *Trichoderma* coiled around the pathogen (Figure 2a) and formed appressoria like structures (Figure 2b). As observed with the strains #5001, the strain 5029 also used the similar mycoparasitic structures against *P. noxium*. Moreover, surface colonisation by *Trichoderma* on the host mycelia was also observed under the microscope (Figure 2c).

### 3.3. Detection of the Antagonistic Activities of Trichoderma Strains against P. noxium in Dual Culture Assay

The results show that the *Trichoderma* strains used in this study were effective in inhibiting *P. noxium* growth (Appendix A). The growth inhibition caused by *Trichoderma* strains was significantly different between *P. noxium* strains (*F* = 7.493, *p* < 0.001) (Figure 3a). The growth inhibition caused by the strains #5001 on *P. noxium* was significantly (*p* = 0.04) higher than the inhibition caused by the strain #5029. The growth inhibition caused by the strain #5001 on *P. noxium* strain B was significantly higher (*F* = 15.391, *p* < 0.001) than *P. noxium* A (62%), D (60%), E (60%) and F (60%). Whereas the inhibition caused by the strain #5029 on *P. noxium* strains B and A was significantly higher (*F* = 17.7, *p* < 0.001) than the strains E (67%), F (65%) and minimum in D (61%) (Figure 3a).

### 3.4. Inhibitory Effect of Volatile Compounds of Trichoderma Strains on P. noxium Growth

The results show that *P. noxium* growth was inhibited by the volatile compounds emitted by the *Trichoderma* strains. The growth inhibition caused by the volatile compounds of *Trichoderma* strains was significantly different between *P. noxium* strains (*F* = 3.104, *p* = 0.033) (Figure 3b). The inhibition caused by the strain #5001 was not significantly different (*p* = 0.924) from the inhibition caused by the strain #5029. The growth inhibition caused by the VOCs of the strain #5001 on *P. noxium* strains B and E was significantly higher (*F* = 4.886, *p* = 0.019) than the inhibition on *P. noxium* strains A (58%), D (53%) and minimum in F (49%). Whereas the inhibition caused by the VOCs of the strain #5029 on *P. noxium* strain B was significantly higher (*F* = 5.99, *p* = 0.010) than strain F (56.9%), D (56.12%), A (55.2%) and minimum in E (52.9%) (Figure 3b).

### 3.5. Changes in the Volatile Compounds Produced by Trichoderma Strains before and after the Confrontation with P. noxium

The volatile compounds produced by strain #5001 before and after confrontation with *P. noxium* strain B are shown in Table 2 and Figure 4, whereas volatile compounds produced by strain #5029 before and after confrontation with *P. noxium* strain B are shown in Table 2 and Figure 5. The compounds identified in pure cultures of *Trichoderma* strains and in co-culture treatments were members of the compound classes; alcohol, aldehyde, ester, diterpenoid alcohol and sesquiterpene (Table 2). A total of 21 compounds were tentatively identified using the NIST library for the pure culture of strain #5001 (Table 2), whereas 19 compounds were identified in co-culture set up of strain #5001–*P. noxium*. On the other hand, a total of 19 compounds were identified for the pure culture of strain # 5029, while 18 compounds were identified in co-culture set up of strain #5029-*P. noxium*. The volatile profiles of the strain #5001 and co-culture of the strain #5001–*P. noxium* were, nearly the same: 16 of 21 compounds were in common. Similarly, 15 of 19 compounds from the strain #5029 were detected in the emission profile of co-culture of strain #5029–*P. noxium*. On the other hand 13 out of 16 compounds seen in a pure *P. noxium* culture, furan, β- necoclovene, aristolene, thujopsene, 2-isopropenyl-4a,8,dimethyl-1,2,3,4a,5,6,7-octahydronaphthalene, δ-gurjunene, β-elemene, β-caryophyllene, 8-isopropenyl-1,5-dimethyl-cyclodeca-1,5-diene, cadiene, elixene and a unidentified sesquiterpene (Appendix A) were not detected in the co-cultures of *P. noxium*- strain #5001 or in the co-cultures of *P. noxium*—strain #5029. Overall, co-culture profiles showed high similarity with the profile of the *Trichoderma* strains rather than of *P. noxium* (Table 2 and Figure 4 and Figure 5). This confirmed the inhibitory effect of *Trichoderma* strains on *P. noxium* growth in the co-culture set up.

The most dominant volatile compound produced by the pure cultures of strain # 5001 and #5029 was ethanol, at a retention time of 1.02 min with relative percent area of 22.72% and 31.18%, respectively, (Table 2, Figure 4 and Figure 5), While, the relative percentage area of ethanol was increased to 33.98% and 36.3%, respectively in the co-culture setup of the strain #5001–*P. noxium* and strain #5029–*P. noxium*. The second most dominant compound produced by the strain #5001 and #5029 was β-cedrene, at 11.72 min with relative percent area of 19.95% and 18.31%, respectively. The emission of β-cedrene was slightly reduced to 19.28% and 16.5% in the co-culture of strain #5001–*P. noxium* and in the co-culture of strain #5029–*P. noxium*, respectively. In addition, isobutanol produced by pure cultures of strains #5001 and #5029, at retention time of 1.33 min with relative percent area of 7.90% and 6.60%, respectively, was also reduced to 4.24% in the co-culture of strain #5001–*P. noxium* and 4.53% in the co-culture of strain #5029–*P. noxium*. There were three new sesquiterpene compounds (α-copaene, β-himachalene, α-longipinene) detected in the co-culture set up of the strain #5001-*P. noxium* and in the cultures of the strain #5029-*P. noxium* were not detected when the strains were grown alone (Table 2, Figure 4 and Figure 5). α-cubebene and acoradiene were unique in the co-culture set up of strain #5001–*P. noxium*, whereas 2-methylbutyl acetate was unique in the co-culture assay of strain #5029–*P. noxium*.

### 3.6. Effect of Diffusible Compounds of Trichoderma Strains on P. noxium Growth

The diffusible compounds emitted by the *Trichoderma* strains showed growth inhibition on *P. noxium* strain B (Figure 3c). The growth inhibition caused by the crude extract was significantly different (*F* = 815.24, *p* < 0.001) among the tested concentration. The inhibition caused by the crude extracts of the strain #5001 was not significantly different from the inhibition by the extracts of strain #5029 (*p* = 0.58). The inhibition caused by the crude extract of the strain #5001 (*F* = 536.84, *p* < 0.001) and #5029 (*F* = 499.85, *p* < 0.001) were significantly high at 10,000 ppm compared to other tested concentrations. Whereas the lowest inhibition by the extracts from both the strains was observed at 500 ppm (Figure 3c).

### 3.7. The Biocontrol-Associated Genes Expression in Trichoderma Strains during the Interaction with P. noxium Strain B

The gene expression levels measured between the strains were significantly different (*p* < 0.04). Expression of *chit* was higher than other biological control related genes expression levels measured in both *Trichoderma* strains (Figure 6). Particularly, relative expression of *chit* gene was increased by 11.6 times in *Trichoderma* strain 5029 and by 8.4 times in *Trichoderma* strain #5001 (Figure 6), While relative expression of *endo*, *bgn*, *pra*, *prb* and *qud* genes were increased by 3.3, 2.5, 3.1 and 2.9 times, respectively in *Trichoderma* strain 5001 and by 1.5, 4.4, 2.4 and 2 times, respectively in *Trichoderma* strain #5029 (Figure 6). The *chit*, *qid* and *bgn* genes of *Trichoderma* strain #5029 were expressed at 1.38, 3.13 and 1.74 times higher, respectively, than expressions detected in *Trichoderma* strain #5001 (Figure 6). Whereas *endo*, *pra* and *prb* genes of the strain #5001 expressed 2.2, 1.3 and 1.4 times higher, respectively, than expression measured in *Trichoderma* strain #5029 (Figure 6).

### 3.8. Effect of Volatiles Produced by Trichoderma Strains on the Growth Parameters of Arabidopsis thaliana

The plants from the shared atmosphere with the *Trichoderma* strains showed improved growth than control plants (Appendix A). The root length of the plants, which shared the space with the strains #5001 and #5029 were significantly (*F* = 1986.4, *p* < 0.001) higher than control plants (Figure 7). Similarly, shoot length (*F* = 3362.7, *p* < 0.001), chlorophyll content (*F* = 41.7, *p* < 0.001) and fresh mass (*F* = 565, *p* < 0.001) were also significantly improved in the plants which shared the space with the strains #5001 and #5029 (Figure 7). The root length and chlorophyll content measured different from control (*F* = 1986.4, *p* < 0.001) but were not significantly different between the treatments of the strain #5001 and #5029 (Figure 7). However, shoot length and fresh mass measured in the treatment of *Trichoderma* strain #5001 was significantly (*F* = 1986.4, *p* < 0.001) higher than the strain # 5029 (Figure 7).

### 3.9. Indole 3-Acetic Acid Production by Trichoderma Strains

Both of the strains used in this study produced IAA in the l-tryptophan supplemented media. The IAA produced by the strain #5029 (6.13 ± 0.31 µg/mL) was significantly higher (*p* = 0.03) than the IAA produced by the strain #5001 (4.49 ± 0.19 µg/mL).

### 3.10. Siderophore Production by Trichoderma Strains

Visible orange halo formation surrounding the *Trichoderma* growth on CAS agar media indicate that both the strains produced siderophores on CAS agar media. Halo zone developed by the strain #5029 (18.33 ± 0.5 cm^2^) was significantly larger (*p* = 0.017) than the strain #5001 (14.2 ± 0.45 cm^2^), indicating that the siderophores production by the strain #5029 was most efficient than the strain #5001 (Figure 8).

## 4. Discussion

Overall results indicate that *Trichoderma* strains used possessed different mechanisms that provided effective antagonism against *Pyrrhoderma noxium*. The *Trichoderma* strains used in this study produced coil around *P. noxium*, while the flattened, enlarged hyphal tips produced by the strains might involve in the direct penetration and ultimately, lead to leakage of cytoplasm and death. Findings also confirm that the strains #5001 and #5029 can grow faster and easily compete for space and nutrients with the slower growing *P. noxium*, which can thus lead to the elimination of the pathogen as previously reported by other researchers [98,99,100,101].

The induced expression of *qid* genes in the strains #5001 and #5029 indicate that *Trichoderma* strains were able to sense the presence of *P. noxium* and prepared for defense by protecting their cell wall from the lytic enzymes of *P. noxium*. Similar observations were also made for *T. harzianum* against *F. solani* and *T. virens* against *F. oxysporum* [23,24]. While induced expression of *chit* and *endo* genes in *Trichoderma* strains #5001 and #5029 indicate that chitinolytic enzymes produced by the strains possibly played a significant role in inhibiting the growth of *P. noxium* (Figure 6). The induced expression of chitinase encoding gene in recombinant *T. harzianum* strain was previously reported to increase its antagonizing activity against *R. solani* [21]. Similarly, expression of chitinase gene (*chi*46) of *Trichoderma reesei* PC-3-7 in *E. coli* showed lytic activity against the cell wall of *Fusarium oxysporum* [102]. In a different study, induced expression of *endo* and *chit* genes were detected in *T. harzianum*, when it was cultured in the medium supplement with inactivated *F. solani* cell wall [26]. Therefore, the induced expression of the genes encoding the chitinase and endochitinases in the strains #5001 and #5029 may be associated with degradation of chitin in the cell wall of *P. noxium*.

Over expression of *prb* and *pra* in the strains #5001 and #5029 indicate that proteinases may play an important role in biological control by the *Trichoderma* strains used in the study (Figure 6). Induced expression of subtilisin-like protease prb1 gene was previously detected in *T. harzianum* transformants during the confrontation with *Rhizoctonia solani* and also in the presences of cell walls of *R. solani* [29]. Moreover, in a greenhouse study, these transformants significantly reduced the disease caused by *R. solani* in cotton plants [29]. Similarly, the expression of trypsin-like protease pra1 gene was previously reported in *T. atroviride* during physical contact with *R. solani* [103]. Therefore, induced expression of the gene encoding the proteases in the strains #5001 and #5029 may degrade the protein components in the cell wall of *P. noxium* and cause cell lysis.

The crude extracts of *Trichoderma* strains #5001 and #5029 were effective in inhibiting the growth of *P. noxium* (Figure 3c). However, further studies are underway to identify the antifungal metabolites present in the crude extracts of the strains #5001 and #5029. In this study, the VOCs of the strains #5001 and #5029 were also efficient in inhibiting the growth of *P. noxium* (Figure 3b), therefore these strains may be able to be used as a biofumigation source to manage *P. noxium* over the application of toxic fungicides. Similarly, an inhibitory effect by the VOCs of *T. atroviride*, *T. harzianum* and *T. virens* on *P. noxium* was reported by Schwarze et al. 2012 [5] and Burcham et al., 2017 [104]. Inhibitory effects of VOCs of *T. reesei* on *Colletotrichum capsici*, *Aspergillus flavus*, *A. niger*, *Cladosporium cucumerium*, *Fusarium solani*, *F. udum*, *Rhizoctonia solani*, *Rhizoctonia bataticola*, *Rhizopus stolonifera* and *Pythium* species have also been observed by several researchers [105,106].

The VOC profiles of the strains #5001 and #5029 were dominated by sesquiterpenes. These results were also aligned with Guo, (2020) study [107], where 52.69% of the volatile profile of *T. reesei* QM6a was dominated by sesquiterpenes. Sesquiterpenes compounds are lipophilic compounds, therefore can cause osmotic control failure in cell membranes of competitors [108]. On the other hand, they can also act as a solvent to facilitate the transportation of toxic compounds through the membrane into the cell [108]. It remains to be determined if any of these sesquiterpenes inhibit *P. noxium*. However, sesquiterpenes from a number of *Trichoderma* species have been reported to have antibacterial and antifungal activities [109,110,111]. The second most dominant VOC detected in this study was β–cedrene and this compound produced by *T. longibrachiatum* was reported for antifungal activity against *C. lagrnarium* and *B. cinerea* [112,113]. Cedrene production by *T. harzianum*, *T. hamatum*, *T. reesei* and *T. velutinum* was also documented by researchers [107,108,114]. The β–cedrene produced by the strains # 5001 and # 5029 may have antifungal properties, and this finding is worthy of further studies. The 3-methyl-1-butanol, 2-methyl-1-propanol, ethanol and ester compound detected in this this study were also produced by various *Trichoderma* species and reported to have antimicrobial activity [54,55,58,115,116,117,118].

The lipophilic VOCs such as 3-methyl-1-butanol and 2-methyl-1-propanol have high affinity for plasma membrane, therefore they are more toxic to the microorganisms than ethanol [119]. The 3-methyl-1-butanol and 2-methyl-1-propanol from *T. atroviride* were reported to have antifungal activity against *P. infestans* [55]. Ethanol detected in the volatile profile of *T. asperellum*, *T. aggressivum*, *T. longibrachiatum*, *T. pseudokoningiihave*, *T. stromaticum*, *T. virens*, *T. atroviride* and *T. pseudokoningiihave* was reported for antifungal activity at high concentrations [54,55,58,117,118].

Compared to monocultures, co-cultivation settings were found to induce specific changes in the fungal VOC emissions, indicating that there is an interaction between *P. noxium* and *Trichoderma* strains. Similar results were observed during the interaction between *T. hamatum* and *Laccaria bicolor* [114]. Altered VOC profiles were also observed during the interaction between *A. alternata* with *F. oxysporum* and *L. bicolor* with *T. harzianum*, *T. hamatum* and *T. velutinum* [114,120]. Production of new VOCs and concentration changes in co-culture settings may be required for the inhibition of a competitor. 13 out of 16 compounds seen in a pure *P. noxium* culture (Appendix A) were not detected in the co-cultures of strain #5001–*P. noxium* or in the co-cultures of strain #5029–*P. noxium*. Overall, co-culture profiles showed high similarity with the profile of the *Trichoderma* strains rather than of *P. noxium* (Table 2 and Figure 4 and Figure 5). This confirmed the inhibitory effect of *Trichoderma* strains on the growth of *P. noxium* growth in the co-culture set up and also the inhibitory effect on the production of *P. noxium*’s volatile compounds. Therefore, the new compounds detected in the co-culture set were high, likely produced by *Trichoderma* strains.

In addition to the antifungal property, VOCs of *Trichoderma* species were also reported to be associated with plant growth promotion [50,51,52,53,54]. This study clearly shows that VOCs from the strains #5001 and #5029 were effective in increasing the plant biomass, chlorophyll content, shoot and root length (Figure 7). Similar results were reported for the VOCs of *T. atroviride*, *T. virens* and *T. asperellum* on growth parameters of *A. thaliana* [60] and *T. viride* on tomato seedlings [54]. The results obtained in this study contrast the results reported by Nieto-Jacobo et al. 2017 [60], where VOCs of *T. reesei* did not show any effects on plants. This contrasting result may be due to the shorter volatile exposure period, the media used for growing *Trichoderma* (MS media) and strain specificity.

Microbial VOCs including sesquiterpenes, 6-pentyl-α-pyrone (6-PP), 3-methyl-1-butanol and 2-methyl-1-butanol were previously reported to be associated with plant growth promotion [50,51,52,53]. Cedrene from *T. guizhouense* was associated with lateral root formation and primary root elongation in *A. thaliana* [121]. Similarly, 6-PP from *T. atroviride* was reported to induce the lateral root formation in *A. thaliana* [122]. The production of 3-methyl-1-butanol and 2-methyl-1-butanol were also reported to improve plant size and chlorophyll concentration [50,52]. During this study, 6-PP was not detected, therefore, possibly 3-methyl-1-butanol, 2-methyl-1-butanol and sesquiterpenes mainly α–cedrene produced by *Trichoderma* strains #5001 and #5029 were involved in the plant promotion observed in *A. thaliana*.

The *Trichoderma* strains used in this study also produced siderophores. Therefore, if these strains were used in the environmental applications they might possibly compete and suppress the growth of *P. noxium* via rapidly acquired available iron before the pathogen. Furthermore, siderophores producing *Trichoderma* species were also reported to help the transformation of insoluble Fe^3+^ to plant absorbable Fe^2+^ and boosts the plant uptake [37,63,64], while, the IAA produced by the *Trichoderma* strains #5001 and #5029 may asset the recovery of *P. noxium* infected trees via modifying the root architecture, resulting in development of new roots which aid the efficient nutrient and water uptake by the plant

## 5. Conclusions

Mycoparasitic ability of *Trichoderma* strains #5001 and #5029 against *P. noxium* was confirmed and the results showed that during the initial stage of interaction *Trichoderma* hyphae often coil around the *P. noxium* and attach to the host via appressorium like structure. After these mycelial interactions, the genes related to the production of cell wall degrading enzymes are expressed and facilitate the penetration of *Trichoderma* hyphae through the *P. noxium* cell wall. Similarly, the volatile and diffusible compounds produced by *Trichoderma* strains also played significant roles in inhibiting the growth of *P. noxium*. It is conceivable that, during the interaction between *Trichoderma* and *P. noxium*, all these mechanisms may be operative simultaneously in order to inhibit the growth of the pathogen. The ability of *Trichoderma* strains to produce IAA and siderophores indicate the possibility of these biological control agents to promote plant growth in environmental applications. The overall study confirmed that *Trichoderma* strains can be used as potential biological control agents against *P. noxium* in environmental applications especially for protection of heritage fig trees.

## Figures and Tables

**Figure 1 jof-08-01105-f001:**
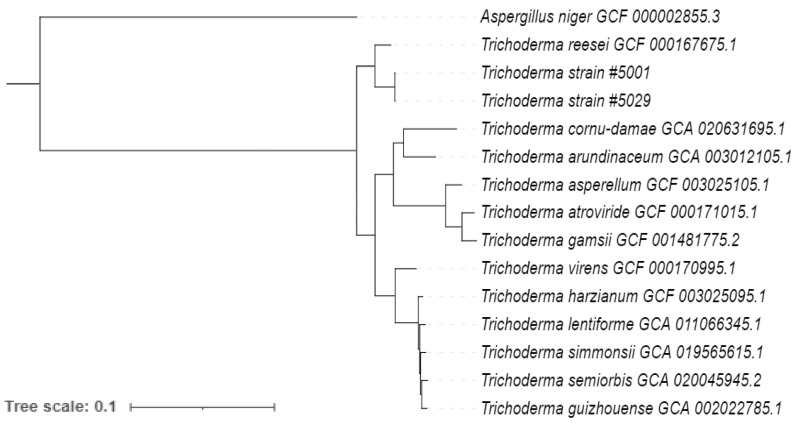
Maximum-likelihood phylogenetic tree of 15 *Trichoderma* strains constructed using 3059 single-copy genes present in all the analysed genomes, using the maximum-likelihood method and with *A. niger* as the outgroup.

**Figure 2 jof-08-01105-f002:**
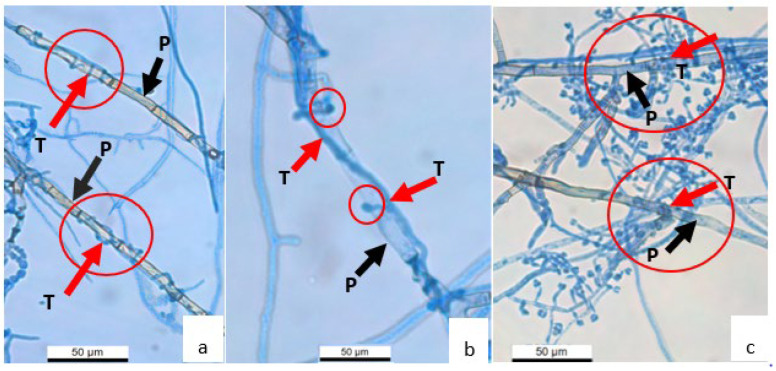
Mycoparasitism by the *Trichoderma* strain #5001 (T) on *P. noxium* strain B (P). (**a**) *Trichoderma coiled* around hyphae of *P. noxium* strain B (X 400), (**b**) appressorium-like structure (X 400) and (**c**) *Trichoderma* growth over *P. noxium* strain B hyphae (X 400). The red and black arrows indicate the *Trichoderma* and *P. noxium*, respectively. Circles indicate the coiling, appressorium-like structures and *Trichoderma* overgrowth on *P. noxium*.

**Figure 3 jof-08-01105-f003:**
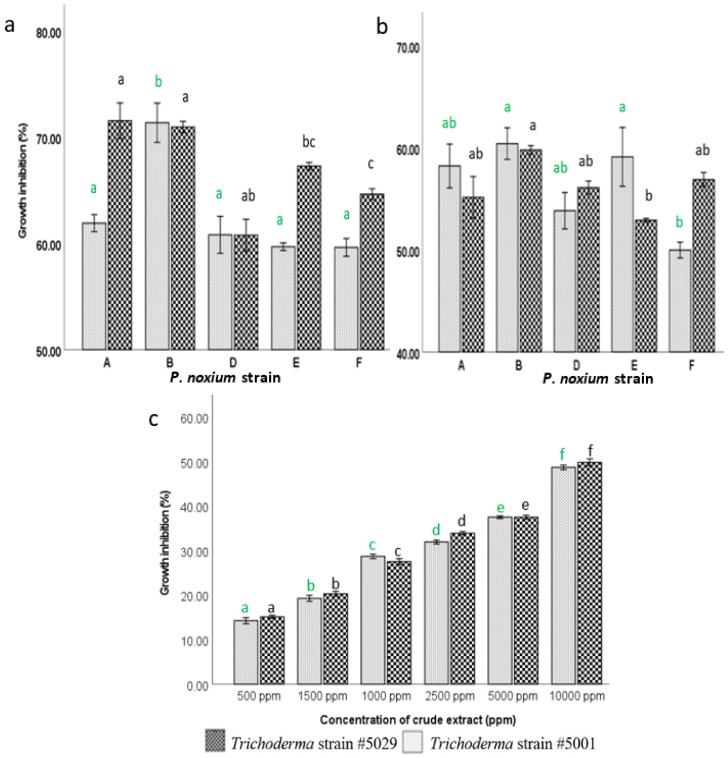
Growth inhibition % of *P. noxium* (strains: A, B, C, D, E and F) (**a**) during the confrontation with *Trichoderma* strains #5001 and #5029 in dual culture set up; (**b**) by the volatile compounds of *Trichoderma* strains #5001 and #5029; (**c**) growth inhibition % of *P. noxium* strain B by the crude extracts of the *Trichoderma* strains #5029 and #5001 at different concentrations. Different letters with the same color on the top of SE bars indicate significant differences in growth inhibition observed between *P. noxium* strains caused by the particular *Trichoderma* strain based on Tukey’s test, *p* < 0.05.

**Figure 4 jof-08-01105-f004:**
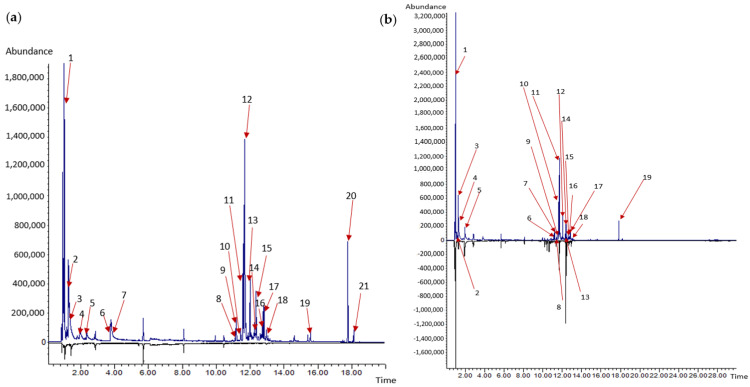
Overlay plot of (**a**) pure culture of strain #5001 on top and media control at bottom; (**b**) co−culture of strain #5001 with *P. noxium* on top and pure culture of *P. noxium* strain at bottom. Numbers refer to compounds listed in Table 2.

**Figure 5 jof-08-01105-f005:**
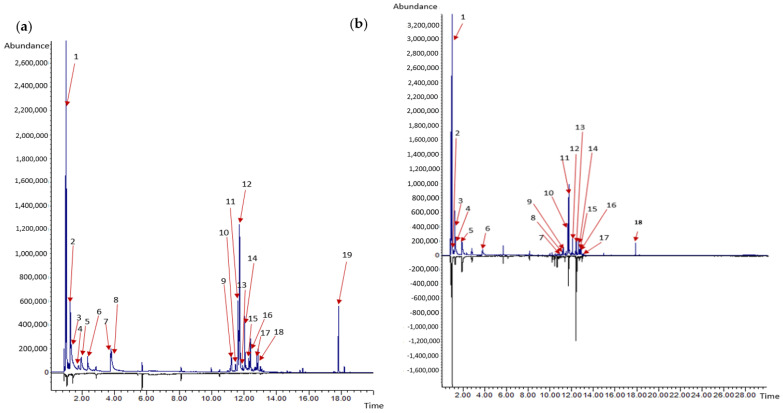
Overlay plot of (**a**) pure culture of strain #5029 on top and media control at bottom; (**b**) co−culture of strain #5029 with *P. noxium* on top and pure culture of *P. noxium* strain at bottom. Numbers refer to compounds listed in Table 2.

**Figure 6 jof-08-01105-f006:**
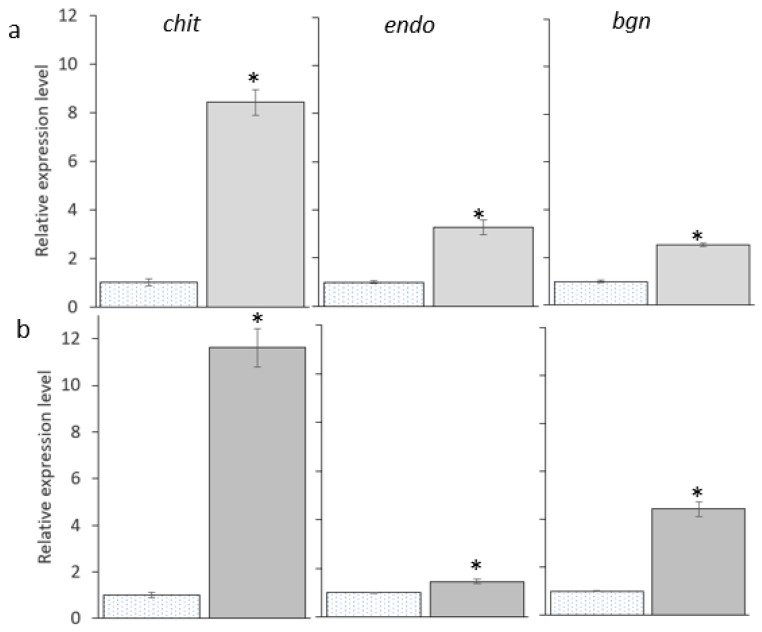
Expression patterns of biocontrol-associated genes after confrontation with *P. noxium* strain B in (**a**,**c**) *Trichoderma* strain #5001 and (**b**,**d**) *Trichoderma* strain #5029. Elongation factor 1−α and β−actin genes (*tef* and *act*, respectively) were used as references to normalize biological control associated genes expression in *Trichoderma* strain #5001 and #5029. Asterisks indicate significant differences by Student’s *t*-test (*p* < 0.05).

**Figure 7 jof-08-01105-f007:**
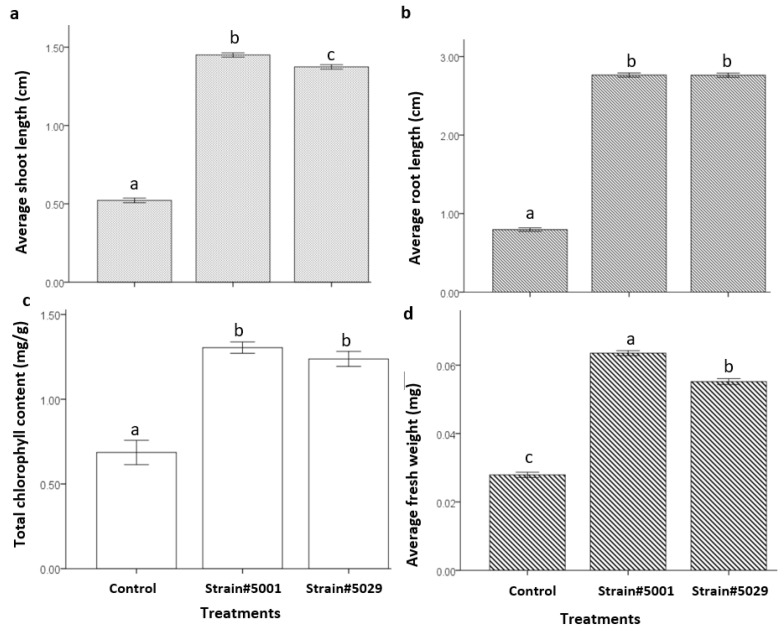
Effect of volatiles produced by the *Trichoderma* strains on plant growth parameters: (**a**) shoot length (cm), (**b**) root length, (**c**) total chlorophyll content (mg/g) and (**d**) fresh weight. Different letters on the top of SE bars indicate significant differences between the treatments based on Tukey’s test, *p* < 0.05.

**Figure 8 jof-08-01105-f008:**
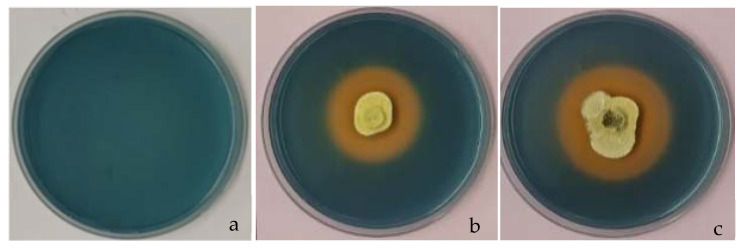
Production of siderophores screened on CAS agar medium (front view) (**a**) uninoculated CAS agar medium, (**b**) CAS medium inoculated with the strain #5001, (**c**) CAS medium inoculated with the strain #5029.

**Table 1 jof-08-01105-t001:** Comparison of genome assembly statistics of *Trichoderma* strain #5001 and #5029 with the reference strain *Trichoderma reesei*.

Genome Features	*Trichoderma* Strain #5001	*Trichoderma* Strain #5029	*Trichoderma reesei* (PRJNA15571)
Total scaffold length, Mb	32.126	32.132	33.396
Total contig length, Mb	32.124	32.13	33.348
Number of contigs (Total)	162	147	128
Number of scaffold (Total)	116	85	77
Largest contig, Kb	964.461	981.4	2199
Largest scaffold, Mb	1.473	1.76	3.757
Scaffold N_90_, KB	196.64	264.963	407.093
Scaffold N_50_, KB	493.501	599.273	1220
GC (%)	53.33	53.33	52.82
Busco completeness (%)	99	99.06	98.3
Total protein coding genes	8812	8969	9111

**Table 2 jof-08-01105-t002:** Changes in the volatile compounds produced by *Trichoderma* strains before and after the confrontation with *P. noxium* tentatively identified through SPEM GC/MS analysis.

		*Trichoderma* Strain #5001			Co-Culture of *Trichoderma* Strain #5001–*P. noxium* Strain B			*Trichoderma* Strain #5029			Co-Culture of *Trichoderma* Strain #5029–*P. noxium* Strain B		
Compound	Class	Retention Time	Area %	Peak	Retention Time	Area %	Peak	Retention Time	Area %	Peak	Retention Time	Area %	Peak
Ethanol	alcohol	1.02	22.72	1	1.02	33.98	1	1.02	31.2	1	1.01	36.3	1
Ethyl acetate	ester	1.28	6.15	2	1.28	6.13	3	1.27	5.01	3	1.26	6.47	3
Isobutanol	alcohol	1.33	7.9	3	1.33	4.24	4	1.32	6.6	4	1.31	4.53	4
Isopentanol	alcohol	1.96	1.76	4	1.95	4.98	5	1.94	2.42	5	1.93	5.85	5
Un known sesquiterpene	sesquiterpene	11.22	2.19	8	11.2	2.48	7	11.2	2.04	9	11.2	2.02	8
α-cedrene	sesquiterpene	11.48	0.94	9	11.5	1.08	8	11.5	0.67	10	11.5	0.83	9
τ-muurolene	sesquiterpene	11.63	7.43	10	11.6	7.63	9	11.6	6.5	11	11.6	6.43	10
β-cedrene	sesquiterpene	11.72	19.95	11	11.7	19.28	10	11.72	18.31	12	11.7	16.5	11
Eremophilene	sesquiterpene	12.05	4.14	13	12.1	3.8	12	12.1	4.12	14	12.1	2.56	12
Di-epi- α -cedrene	sesquiterpene	12.4	3.17	15	12.4	2.85	14	12.4	2.77	16	12.4	2.31	13
Cuparene	sesquiterpene	12.79	1.8	16	12.8	1.39	17	12.8	1.29	17	12.8	1.15	16
δ-elemene	sesquiterpene	12.87	2.16	17	12.9	1.68	18	12.9	1.71	18	12.9	1.37	17
Verticiol	diterpenoid alcohol	17.83	5.91	20	17.8	3.01	19	17.8	4.92	19	17.8	1.74	18
α-cubebene	sesquiterpene	11.78	1.39	12	11.8	0.77	11	11.8	1.22	13	-	-	-
Acoradiene	sesquiterpene	12.3	1.44	14	12.3	0.93	13	12.3	0.85	15	-	-	-
Isobutyl acetate	ester	2.34	1.22	5	-	-	-	2.33	2.21	5	-	-	-
Isopentyl acetate	ester	3.78	0.89	6	-	-	-	3.77	0.95	7	-	-	-
2-methylbutyl acetate	ester	3.81	3.27	7	-	-	-	3.8	4.72	8	3.82	2.23	6
Isobutanal	aldehyde	-	-	-	1.16	1	2	1.15	0.85	2	1.14	0.78	2
α-copaene	sesquiterpene	-	-	-	11.2	0.75	6	-	-	-	11.1	0.63	7
β-himachalene	sesquiterpene	-	-	-	12.6	0.7	15	-	-	-	12.6	2.68	14
α-longipinene	sesquiterpene	-	-	-	12.7	0.5	16	-	-	-	12.7	1.21	15
Un known diterpene	diterpene	18.19	0.71	21	-	-	-	-	-	-	-	-	-
δ-cadinene	sesquiterpene	13.01	0.89	18	-	-	-	-	-	-	-	-	-
α-cubebene	sesquiterpenes	15.62	0.68	19	-	-	-	-	-	-	-	-	-

## Data Availability

Data are provided in the Appendix A. Genome data of *Trichoderma* strain #5001 (SAMN30380231) and #5029 (SAMN30380232) were deposited in NCBI BioProject under reference number: PRJNA870600. ITS sequences of *P. noxium* strain A, B, D, E and F were deposited under GenBank accession numbers: OP430849, OP430850, OP430851, OP430852 and OP430853, respectively.

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
