# Peer review of "Assessing the Various Antagonistic Mechanisms of Trichoderma Strains against the Brown Root Rot Pathogen Pyrrhoderma noxium Infecting Heritage Fig Trees"

_jof, 2022, doi:10.3390/jof8101105_

Round 1
Reviewer 1 Report
This MS describes two Trichoderma strains (#5001 and #5029) to protect trees against Pyrrhoderma noxium. The author studied mycoparasitism mechanism, antifungal compounds and beneficial prospects of these two strains, which maybe developed as bicontrol agents. I recommend mino revision for this article, but following comments should be addressed.
I would suggest authors to identify the species of these two two Trichoderma strains (#5001 and #5029) and the sequences should be submitted to NCBI. The authors should also provide reference about Pyrrhoderma noxium strain A, B, D, E and F used in this article.
The following specific points could be addressed.
Figure 1b. Why is the P. noxium hypha morphology in Figure 1b such as color and transparency different from Fig 1a and c.?
And Fig1 only show strain #5001. #5029 behavior should also be added.
FigA1, which P. noxium strain shoud be noted.
In other experiments including Fig 3, add P. noxium strain name
Table A2, I didn’t see highlighted compounds names as footnote indicated.
The same question in Table A3
Line 423 and 433, the compound table produced by a pure P. noxium culture was not listed in the paper. you can list as a supplementary table.
The author only describe the effects of volatiles on plant growth. Add results of diffusible compounds effects on plant growth.
Author Response
REVIEWER #1
This MS describes two Trichoderma strains (#5001 and #5029) to protect trees against Pyrrhoderma noxium. The author studied mycoparasitism mechanism, antifungal compounds and beneficial prospects of these two strains, which maybe developed as bicontrol agents. I recommend mino revision for this article, but following comments should be addressed.
I would suggest authors to identify the species of these two two Trichoderma strains (#5001 and #5029) and the sequences should be submitted to NCBI. The authors should also provide reference about Pyrrhoderma noxium strain A, B, D, E and F used in this article.
Sequences are now submitted and deposition information is provided in the text
The following specific points could be addressed.
- Figure 1b. Why is the noxium hypha morphology in Figure 1b such as color and transparency different from Fig 1a and c.?
Microscopic photos were taken from the same specimen at same magnification, Figure 1b was taken while focusing the attachment of appressorium-like structure of strain #5001 on P. noxium. That could be the reason color and transparency of hypha morphology of P. noxium bit different. However, we can clearly see the appressorium-like structure of strain #5001 on P noxium. it is an artefact of photography.
- And Fig1 only show strain #5001. #5029 behavior should also be added.
Strain #5001 and 5029 were identified as closely related to T. reesei, both showed same mycoparasitism against P. noxium. Therefore, # 5001 was used as representative strain in Figure 1.
- FigA1, which noxium strain shoud be noted.
Strain information of P. noxium is provided in the figure 1
- In other experiments including Fig 3, add noxium strain name
Strain information of P. noxium is provided in the figure 1
- Table A2, I didn’t see highlighted compounds names as footnote indicated.
Table modified and A2 and A3 combined to show the changes in volatile compounds production in the pure cultures of Trichoderma and co-culture Trichoderma with P. noxium. Footnote is removed.
- The same question in Table A3
Table modified and A2 and A3 combined to show the changes in volatile compounds production in the pure cultures of Trichoderma and co-culture Trichoderma with P. noxium. Footnote is removed.
- Line 423 and 433, the compound table produced by a pure noxium culture was not listed in the paper. you can list as a supplementary table.
Compounds produced by pure culture of P. noxium culture is included as a supplementary table.
- The author only describe the effects of volatiles on plant growth. Add results of diffusible compounds effects on plant growth.
This paper only focused on the effect of volatile compounds on plant growth. Effect of diffusible compounds on plant growth will follow as a forthcoming manuscript.
Reviewer 2 Report
Authors comprehensively evaluated the antifungal activities of two Trichoderma strains #5001 and #5029 against Pyrrhoderma noxium and revealed their potential in biocontrol of Brown root rot. However, the manuscript needs to be largely modified. The main comments are as follows:
1. Identification of both Trichoderma strains must be provided before biocontrol assays were performed.
2. The introduction is generic and needs to focus on the biocontrol of the target pathogen/disease. Advances in metabolites produced by Trichoderma species and their genes related to biological control need to be supplemented in this section. The significant of this study is not emphasized. However, excess background and researches are presented in Discussion, which should be moved to the section of Introduction.
3. I would suggest to re-organize the manuscript according to the effects of the biocontrol fungi, i.e. mycoparasitism-antagonistic activities and compounds-growth promotion and related metabolites. Extracellular lytic enzymes are more likely to be regarded to involve in mycoparasitic process via degrading hosts’ cell walls and facilitating the penetration of the mycoparasites. The division of volatile and diffusible compounds is fine; however, antagonistic activities are synthetic effects, and emphasis of two kinds of compounds might not quite match the title of the paper.
4. P83, “The mycoparasitic activity of Trichoderma strains” was not correct.
5. Appended figures and tables should be separated from the main text, otherwise the structure of the whole paper looks confused.
6. Figure 1: The arrows in Fig.1b are ambiguous. Fig.1c just shows the space position of both fungi, but cannot reflect the association (“Trichoderma growth over P. noxium hyphae”) between them. The notes of the arrows should be provided in the legend.
7. Figure 2 and 4: The effects of Trichoderma stain #5001 and #5029 to different P. noxium isolates should be grouped instead of comparing the differences of two biocontrol isolates. Moreover, ID numbers of the six pathogenic strains should be given, and a term with singular form is used in coordinate axis.
8. Figure 3 and 7 are ambiguous, graphs with high resolution and stark contrasts in color should be adopted.
9. Figure 5 and 6: Change the title to “Volatile compounds produced by Trichoderma strains during confrontation with P. noxium”, then provide the experimental methods below. Importantly, it is not accurate to cite the information from appended files (Table A2 and A3). In addition, the graphs need to be greatly lengthened to show the peaks clearly.
10. Figure 8: Similar to Figure 2 and 4.
11. Cancel Figure 9, result description is clear.
12. Table A1 is not accepted using the staggered arrangement of the column and contents.
13. Appended figures should be arranged in order. Adjust the sequence of the notes in Figure A1 according to lowercase letters. Delete Figure A2, for the readers can grasp the idea from Figure A3.
14. It’s more interesting and informative to detect the changes of the volatile components and their contents with and without stress, so I would suggest to assess the compounds produced in pure culture of a Trichoderma strain and in co-culture with the pathogens instead of comparing the different isolates (Table A2 and Table A3).
15. P546-548, “Over expression of prb and pra in the strains--- (Figures 8 and 9)” is not touched in this paper.
16. As a research article, too many references are cited. The references should be checked carefully to avoid mistakes. For example, the titles of ref. 14, 19, 20, 21, 29 are case sensitivity. In ref. 15 and 20, normalized abbreviated journal names should be adopted. Some information, e.g. ref. 20 and 29, are imperfect.
Author Response
REVIEWER #2
Authors comprehensively evaluated the antifungal activities of two Trichoderma strains #5001 and #5029 against Pyrrhoderma noxium and revealed their potential in biocontrol of Brown root rot. However, the manuscript needs to be largely modified. The main comments are as follows:
- Identification of both Trichoderma strains must be provided before biocontrol assays were performed.
identification of Trichoderma strains included.
- The introduction is generic and needs to focus on the biocontrol of the target pathogen/disease. Advances in metabolites produced by Trichoderma species and their genes related to biological control need to be supplemented in this section. The significant of this study is not emphasized. However, excess background and researches are presented in Discussion, which should be moved to the section of Introduction.
Introduction and discussion sections were re organized as per reviewer’s comment.
- I would suggest to re-organize the manuscript according to the effects of the biocontrol fungi, i.e. mycoparasitism-antagonistic activities and compounds-growth promotion and related metabolites. Extracellular lytic enzymes are more likely to be regarded to involve in mycoparasitic process via degrading hosts’ cell walls and facilitating the penetration of the mycoparasites. The division of volatile and diffusible compounds is fine; however, antagonistic activities are synthetic effects, and emphasis of two kinds of compounds might not quite match the title of the paper.
Method and result section rearranged as suggested by reviewer.
Title has changed accordingly
- P83, “The mycoparasitic activity of Trichoderma strains” was not correct.
Mycoparasitic activity is changed to antagonistic activity.
- Appended figures and tables should be separated from the main text, otherwise the structure of the whole paper looks
Appended figures and tables are separated from main text and provided at the end.
- Figure 1: The arrows in Fig.1b are ambiguous. Fig.1c just shows the space position of both fungi, but cannot reflect the association (“Trichoderma growth over P. noxium hyphae”) between them. The notes of the arrows should be provided in the legend.
Positions of the arrows readjusted. Footnote provided to explain the arrows and circles.
- Figure 2 and 4: The effects of Trichoderma stain #5001 and #5029 to different P. noxium isolates should be grouped instead of comparing the differences of two biocontrol isolates. Moreover, ID numbers of the six pathogenic strains should be given, and a term with singular form is used in coordinate axis.
Figure 2, 4 and 8 were grouped. Singular form strain is used in X axis
- Figure 3 and 7 are ambiguous, graphs with high resolution and stark contrasts in color should be adopted.
Figure 3 and 7 graphs are regenerated with high resolution and stark contrast colors used.
- Figure 5 and 6: Change the title to “Volatile compounds produced by Trichoderma strains during confrontation with P. noxium”, then provide the experimental methods below. Importantly, it is not accurate to cite the information from appended files (Table A2 and A3). In addition, the graphs need to be greatly lengthened to show the peaks clearly.
Chromatograms were lengthened to clearly show the peaks.
- Figure 8: Similar to Figure 2 and 4.
Figure 2, 4 and 8 were grouped. Singular form strain is used in X axis
- Cancel Figure 9, result description is clear.
Figure 9 removed
- Table A1 is not accepted using the staggered arrangement of the column and contents.
Table formatted
- Appended figures should be arranged in order. Adjust the sequence of the notes in Figure A1 according to lowercase letters. Delete Figure A2, for the readers can grasp the idea from Figure A3.
Appended figures arranged in order.
Figure 2 deleted
- It’s more interesting and informative to detect the changes of the volatile components and their contents with and without stress, so I would suggest to assess the compounds produced in pure culture of a Trichoderma strain and in co-culture with the pathogens instead of comparing the different isolates (Table A2 and Table A3).
As per reviewers comment Table A2 and A3 combined and comparisons were made between the volatile compounds produced in the pure cultures of Trichoderma and co-cultured Trichoderma with P. noxium strain B.
- P546-548, “Over expression of prb and pra in the strains--- (Figures 8 and 9)” is not touched in this paper.
Expression of prb and pra were described in result and discussion section.
- As a research article, too many references are cited. The references should be checked carefully to avoid mistakes. For example, the titles of ref. 14, 19, 20, 21, 29 are case sensitivity. In ref. 15 and 20, normalized abbreviated journal names should be adopted. Some information, e.g. ref. 20 and 29, are imperfect.
Changes made in the reference section as indicated by reviewer.
Reviewer 3 Report
1. Line 91. The authors need to clearly describe how they determine growth inhibition percentage, rather than making the audience refer to the literature.
2. Line 326. The letters in Figure 2 are too small for color recognition. The authors need to revise the figure.
3. Line 355. The letters in Figure 3 are too small to be recognized. The authors need to revise the figure.
4. Line 372. Again, the letters in Figure 4 are too small for color recognition. The authors need to revise the figure.
5. Again, the letters in Figure 5, 6, 7, 8 and 9 are too small to be recognized. The authors need to revise these figures.
6. The authors collected the VOCs by co-culturing #5001 and #5029 separately with P. noxium, and they identified several VOCs that are produced only in the co-cultures. The authors believe that these VOCs are produced by #5001 and #5029 following contact with P. noxium. However, they cannot rule out the possibility that these VOCs are produced by P. noxium following contact with Trichoderma mycelia unless dead P. noxium mycelia were used in the co-cultures. The authors need to address this possibility in the Discussion.
7. The description of IAA production in lines 479 and 480 contradicts the results in Figure 9. Besides, the authors mention that “the IAA produced by the strain #5029 (it should be #5001) is 5.28+0.48μg/mL”. However, according to Figure 9, strain #5001 produced IAA higher than 6 μg/mL. The authors need to revise the description.
8. In discussion, when the authors interpret their results, they should refer to the corresponding figures to give the audience an idea what the authors are discussing.
Author Response
REVIEWER #3
- Line 91. The authors need to clearly describe how they determine growth inhibition percentage, rather than making the audience refer to the literature.
Formula is used to describe the how the growth inhibition percentage was calculated.
- Line 326. The letters in Figure 2 are too small for color recognition. The authors need to revise the figure.
Size of the letters on the error bars and figure size are increased.
- Line 355. The letters in Figure 3 are too small to be recognized. The authors need to revise the figure.
Size of the letters on the error bars and figure are size increased.
- Line 372. Again, the letters in Figure 4 are too small for color recognition. The authors need to revise the figure.
Size of the letters on the error bars and figure size are increased.
- Again, the letters in Figure 5, 6, 7, 8 and 9 are too small to be recognized. The authors need to revise these figures.
Size of the letters on the error bars and figure size are increased.
- The authors collected the VOCs by co-culturing #5001 and #5029 separately with noxium, and they identified several VOCs that are produced only in the co-cultures. The authors believe that these VOCs are produced by #5001 and #5029 following contact with P. noxium. However, they cannot rule out the possibility that these VOCs are produced by P. noxium following contact with Trichoderma mycelia unless dead P. noxium mycelia were used in the co-cultures. The authors need to address this possibility in the Discussion.
Below paragraph is added to Discussion section
13 out of 16 compounds seen in a pure P. noxium culture (Figure S2 and Table S2) were not detected in the co-cultures of P. noxium- strain #5001 or in the co-cultures of P. nox-ium-strain #5029. Overall, co-culture profiles showed high similarity with the profile of the Trichoderma strains rather than of P. noxium (Table 2 and Figure 5,6). This confirmed the inhibitory effect of Trichoderma strains on the growth of P. noxium growth in the co-culture set up and also the inhibitory effect on the production of P. noxium’s volatile compounds. Therefore, the new compounds detected in the co-culture set were high, likely produced by Trichoderma strains.
- The description of IAA production in lines 479 and 480 contradicts the results in Figure 9. Besides, the authors mention that “the IAA produced by the strain #5029 (it should be #5001) is 5.28+0.48μg/mL”. However, according to Figure 9, strain #5001 produced IAA higher than 6 μg/mL. The authors need to revise the description.
IAA production by the strains are re calculated and used in the result section
- In discussion, when the authors interpret their results, they should refer to the corresponding figures to give the audience an idea what the authors are discussing.
Figures are referred in the discussion section
Round 2
Reviewer 2 Report
The revised manuscript has been greatly improved.
Minor revision:
1. I would suggest not separating antagonistic activities and different kinds of compounds, for they are more closely related than expression of biocontrol-associated genes in this manuscript.
2. Some figure numbers in the main text are confused. P378-381, “Figure 1a-1c” should be “Figure 2a-2c”. P398 and P435, the numbers are not correct. Check other parts as well.
3. P178, Change “Colony size” to “Colony diameter”.
4. Figure 3, Cancel “Inhibition by” in the icons.
5. P682, “T. ressei” should be “T. reesei”.
Author Response
Response to reviewers’ comments
- I would suggest not separating antagonistic activities and different kinds of compounds, for they are more closely related than expression of biocontrol-associated genes in this manuscript.
As per suggestion all the antagonistic activities, volatile and diffusible compounds were put together. Expression of biological control associated genes separated from above mention sections and introduced at a later stage.
- Some figure numbers in the main text are confused. P378-381, “Figure 1a-1c” should be “Figure 2a-2c”. P398 and P435, the numbers are not correct. Check other parts as well.
Figure numbers in the text are double checked and necessary changes are made.
- P178, Change “Colony size” to “Colony diameter”.
As per suggestion, Colony size is change into Colony diameter.
- Figure 3, Cancel “Inhibition by” in the icons.
As per suggestion, “incubation by” removed and on the Trichoderma strain’s numbers used in the Figure 3
- P682, “T. ressei” should be “T. reesei”.
Spelling mistake corrected
